# Unbiased Phosphoproteome Mining Reveals New Functional Sites of Metabolite-Derived PTMs Involved in MASLD Development

**DOI:** 10.3390/ijms242216172

**Published:** 2023-11-10

**Authors:** Eduardo Moltó, Cristina Pintado, Ruy Andrade Louzada, Ernesto Bernal-Mizrachi, Antonio Andrés, Nilda Gallardo, Elena Bonzon-Kulichenko

**Affiliations:** 1Biochemistry Section, Regional Center for Biomedical Research (CRIB), Faculty of Environmental Sciences and Biochemistry, University of Castilla-La Mancha, Avda. Carlos III s/n, 45071 Toledo, Spain; 2Department of Internal Medicine, Division of Endocrinology, Diabetes and Metabolism, Miller School of Medicine, University of Miami, Miami, FL 33136, USA; 3Biochemistry Section, Regional Center for Biomedical Research (CRIB), Faculty of Sciences and Chemical Technologies, University of Castilla-La Mancha, Avda. Camilo Jose Cela 10, 13071 Ciudad Real, Spain

**Keywords:** post-translational modifications, proteomics, mass spectrometry, metabolic dysfunction-associated steatotic liver disease (MASLD)

## Abstract

Post-translational modifications (PTMs) of proteins are paramount in health and disease. Phosphoproteome analysis by enrichment techniques is becoming increasingly attractive for biomedical research. Recent findings show co-enrichment of other phosphate-containing biologically relevant PTMs, but these results were obtained by closed searches focused on the modifications sought. Open searches are a breakthrough in high-throughput PTM analysis (OS-PTM), identifying practically all PTMs detectable by mass spectrometry, even unknown ones, with their modified sites, in a hypothesis-free and deep manner. Here we reanalyze liver phosphoproteome by OS-PTM, demonstrating its extremely complex nature. We found extensive Lys glycerophosphorylations (pgK), as well as modification with glycerylphosphorylethanolamine on Glu (gpetE) and flavin mononucleotide on His (fmnH). The functionality of these metabolite-derived PTMs is demonstrated during metabolic dysfunction-associated steatotic liver disease (MASLD) development in mice. MASLD elicits specific alterations in pgK, epgE and fmnH in the liver, mainly on glycolytic enzymes and mitochondrial proteins, suggesting an increase in glycolysis and mitochondrial ATP production from the early insulin-resistant stages. Thus, we show new possible mechanisms based on metabolite-derived PTMs leading to intrahepatic lipid accumulation during MASLD development and reinforce phosphoproteome enrichment as a valuable tool with which to study the functional implications of a variety of low-abundant phosphate-containing PTMs in cell physiology.

## 1. Introduction

Protein post-translational modifications (PTMs) are covalent modifications that occur in proteins after their synthesis. PTMs vary in size and nature, from small chemical alterations to the addition of functional groups or even the binding of complete proteins. PTMs can be added and removed enzymatically [1], while others are non-enzymatic in origin [2,3]. They are key regulators of protein structure and function and are paramount in health and disease [4,5,6,7,8]. Protein phosphorylation is one of the most studied PTMs due to its high implication in cell signaling networks. It regulates carbohydrates and lipids metabolism [9], energy production, apoptosis [10] and many other processes.

Mass spectrometry (MS) has no rival for the study of PTMs. Current technical advances allow extremely deep and accurate results, producing millions of spectra and thousands of peptides and proteins identified in a single analysis, making this technique attractive for biomedical research. However, phosphopeptides must be isolated by enrichment techniques prior to MS analysis due to their low abundance in complex biological samples compared to their nonmodified counterparts. These techniques include immobilized metal affinity chromatography (IMAC), titanium dioxide chromatography, phospho-specific antibodies and strong cation exchange chromatography (SCX). Combined with current MS instruments and developments in bioinformatics, they attain overwhelming depths of analysis with a near complete identification of the phosphoproteins present in a sample [11]. Since all these enrichment techniques target the negatively charged phosphate groups in phosphopeptides, recent findings demonstrate that apart from the non-phospho acidic [12], N/Q-rich [13], or N-glyco-peptides [14], other phosphate-containing low abundant and biologically relevant PTMs get coenriched with the phosphopeptides, such as Arg-ADP-ribosylations and phosphoribosylations [15], as well as Lys modifications with pyridoxal phosphate [16] or glycerolphosphate [2]. Findings like these suggest that it is possible to mine the phosphoproteome for potential new phosphate-containing biologically relevant PTMs, the identification and characterization of which would boost our current knowledge about the role of such low abundant and poorly studied PTMs in cellular pathophysiology.

Although the findings mentioned above were profound and identified hundreds of modified sites with a particular PTM, they were obtained by traditional closed database searches (CSs), which are focused on the few PTMs under study. CSs rely on an exact match between the experimental and theoretical masses for precursors and fragments at increasingly narrow tolerances (sub ppm) due to the constant increase in instrument resolution. However, the vast majority of PTMs cause a mass difference with the unmodified peptide in the order of tens of daltons (for example, 79.97 Da for phosphorylation), well above the tolerances used for the search. Thus, the fundamental limitation of this strategy for PTM analysis is that the modifications to be studied must be selected beforehand and all peptides containing modifications of chemical or biological origin not defined a priori in the searches are not identified. As a result, most of the acquired spectra remains unassigned.

The recently developed open search (OS) algorithms for high-throughput PTM analysis (OS-PTM) (Comet-PTM, Taggraph, MSFragger) [17,18,19] constitute a breakthrough in this field. In OS-PTM, the mass tolerance for peptide precursors is increased to hundreds of daltons (Da), allowing peptides with unanticipated modifications to be considered by the algorithm for scoring. At the same time the tolerance for the high-resolution MS/MS fragments is kept in the order of ppm to filter out noise. Afterwards, the OS-PTM algorithm calculates the difference between the mass of the unmodified peptide candidate and the experimental mass of the precursor ion (ΔMass). ΔMass is then iteratively added to each amino acid in the peptide sequence, and the position that yields the best score is selected as the correct match. In this way, OS-PTM not only matches unmodified fragments, but also fishes-out the modified ones without defining these mass shifts a priori. Thus, peptides with unanticipated mass-shifts are scored higher than in CS, the position of the modified site can be determined, and they get through the statistical significance filtering. As a result, OS-PTM matches a big proportion of spectra previously not identified by CS [20], and makes it possible to discover new protein variants and post-translational modifications that were not addressed in the CS. Thus, due to its unbiased nature and depth of analysis, OS-PTM has been used to identify the involvement of previously unknown post-translational modifications in diseases [6,19]. However, OS-PTM analysis is still challenging since many detected modifications are unknown and constitute the “dark matter” of proteome data sets [18].

In this work we apply OS-PTM to reanalyze liver phosphoproteome and unveil novel metabolic dysfunction-associated steatotic liver disease (MASLD)-associated metabolite-derived PTMs coenriched with phosphopeptides. MASLD is a global health problem without an effective early diagnosis or approved drugs [21]. There is an urgent need for further insight into disease mechanism, where PTMs are involved, but little is known [22,23,24]. Here we propose new possible mechanisms behind intrahepatic lipid accumulation during MASLD progression based on metabolite-derived PTMs.

## 2. Results

### 2.1. Unbiased Proteome-Wide OS PTM Analysis of Mouse Liver Phosphoproteome Reveals a Complex Picture

With the aim to uncover new biologically relevant phosphate-containing PTMs in the phosphoproteome, we reinterrogated, through hypothesis-free database searches, the raw MS data from a large-scale phosphoproteomics study of mouse liver [24]. Reanalysis of nearly 1 million fragmentation spectra resulted in the identification of 177,554 peptides from 9065 unique proteins, tripling the number originally reported by traditional closed database search and confirming most of the proteins (Figure 1A, Appendix A). Out of the 6180 new proteins identified by us, the 20 most abundant are presented in Figure 1B and are proteins modified in a way impossible to predict, as observed in the case of the most abundant new protein disulfide isomerase (P09103, P4hb) (Figure 1C). Additionally, as shown in Appendix A, OS-PTM confirmed most (82%) plain peptide sequences identified by traditional closed searches (CS) (Appendix A) and expanded unique peptide identities to a 30%. For the remaining 4829 (18%) CS-only plain peptide sequences, the scores obtained for these spectra were similar or higher in the OS-PTM search (Appendix A). The exception was a population of peptides identified by the CS with more than one preselected modification per peptide (blue dots in Appendix A). For some of those peptides, CS obtained a higher score compared to OS-PTM (blue dots below the dotted diagonal line in Appendix A), since composite mass-sifts are treated as a single delta-mass by the OS-PTM algorithm, which affects the matching of fragment ions containing the individual mass-sifts, decreasing the OS-assigned score. However, since the fraction of CS-only multiply modified peptides with a score higher than in OS-PTM was minor, the impact of this limitation on the general performance of the OS-PTM algorithm seems rather small. These results demonstrate that OS-PTM analysis is as reliable as the CS used by the original authors [24], but much deeper, since it considers all the modifications detectable by MS present in a sample.

The distribution profile of all delta masses (ΔMs) detected in the phosphoproteome showed a complex picture, with 977 ΔM peaks identified (Figure 1D). We then calculated the percentage of the PSM sum from each one of these ΔM peaks from the total amount of PSMs composing the 977 ΔM peaks in the experiment. As expected, the most abundant modifications were mono-phosphorylations (ΔM = 79.96 Da, as well as 80.97 and 81.97 Da, corresponding to the presence of one or two C13, respectively) with 35% of the PTM mass, followed by unmodified peptides (0 Da) with 13%. However, the vast majority of the most abundant ΔM peaks were unknown according to UNIMOD (Figure 1E,F), comprising the “dark matter” of proteomics [18].

To elucidate the chemical nature of unknown ΔMs, we performed a system-wide analysis of the ΔM peak distribution per amino acid residue type and peptide sequence quintile. Figure 1E shows the top 50 ΔMs that make up 77% of the total mass of the phosphoproteome PTMs. The ΔM peaks corresponding to clearly nonmodified peptides (ΔM = 0 Da, as well as 1.00, 2.00 and 3.00 Da, corresponding to the presence of one, two or three C13, respectively, or −17.02 Da corresponding to the loss of ammonium from the N-terminal peptides) were left out. We identified modifications on all 20 amino acids and many residues with different ΔMs, making it evident that such a complexity could never be anticipated by a traditional closed search. As expected, mono-phosphorylated residues were mostly Ser, followed by Thr, although phospho-Tyr were also detected.

This type of system-wide analysis (Figure 1E) evidenced a clear bias towards the specific amino acids of many unknown ΔMs and enabled the identification of those that were combinations of two or more abundant ΔMs, such as the combination of phosphorylations with deamidations (79.97 + 0.98 = 80.95 Da) (Appendix A), or with oxidations mostly in Met (79.97 + 15.99 = 95.96 Da) (Appendix A). For some unknown ΔMs, we at least managed to discriminate between their artefactual or biological origin. Biological modifications are the ones present in the sample before protein extraction and preparation for mass spectrometry measurements, while artefactual modifications are the ones unavoidably introduced during the proteomics benchwork. The latter include over-alkylation of Cys and Met [25,26], Met and Cys oxidation during sample preparation [27] or cyclization and ammonia loss during the peptide fragmentation process in the mass spectrometer from peptides containing N-terminal carbamidomethylated Cys (CAM-Cys) [28]. We observed thousands of peptides with Met oxidation or ammonia loss from peptide N-terminal Cam-Cys (Figure 1E, Appendix A). While Met oxidation was evenly distributed among the sequence quintiles of all peptides bearing this mass shift, CAM-Cys ammonia loss was consistently found in the first quintile of all peptide sequences containing this modification (Appendix A). The latter case prompted us to analyze the sequence quintile distribution of unknown ΔMs. We found many of those occurring over 70% of the time in the first quintile of all peptide sequences containing the given ΔM (Figure 1E). Thus, they were concentrated on the newly generated peptide (but not protein) N-terminal and were considered artefactual. This is the case with the combination of C13 or deamidations with ammonia-loss from carbamidomethylated-Cys or Gln at peptide N-terminal (2 × 0.98 − 17.02 = −15.04 Da) (Appendix A). However, when it comes to N-terminal Met, in most cases it corresponds to the N-terminal of proteins, and the ΔM is biological in nature. For example, −9.06 Da corresponds to the combination of the loss of the N-terminal Met of the protein, followed by acetylation of the newly formed protein’s N-terminal, plus the presence of a phosphorylated residue in the same peptide (−89.03 + 79.97 = −9.06 Da) (Appendix A).

Additionally, we find some ΔMs which are annotated in UNIMOD but have a different nature in the phosphoproteome. This is the case of 229.01 Da, which in UNIMOD appears as pyridoxal phosphate specific for Lys. In this regard, pyridoxal phosphate enrichment by IMAC has been reported [16]. However, we clearly see this ΔM on peptide N-terminal His residues (Figure 1E and Appendix A), and did not observe the pyridoxal-specific [C_8_H_10_NO_5_P] (231 Da) loss from the precursor ion in the fragmentation spectra [29], thus it is not pyridoxal phosphate. We were unable to elucidate the chemical nature of this modification. However, it should contain a phosphate in its structure or be a combination of an unknown ΔM affecting N-terminal His with phosphorylation since the typical loss of [H_3_PO_4_] (98 Da) was observed in the fragmentation spectra. Thus, liver phosphoproteome contains an unknown modification of an artefactual nature with a ΔM very close to the pyridoxal phosphate.

Another case is 13.97 Da specific for Trp (Figure 1E, Appendix A), which in UNIMOD appears as a Trp modification with oxolactone. However, the structure of this ΔM must be different, since oxolactone formation would imply the breakdown of the polypeptide chain, and the resulting ΔM would correspond to the formation of an oxolactone plus the loss of a piece of the peptide sequence [30]. In this regard, three alternative structures for internal Trp residues modified by 13.97 Da were proposed [31].

All new ΔMs identified by us are noted in red in Figure 1E (see Appendix A for the complete list of annotated ΔMs). These system-wide analyses will enable us to enrich the UNIMOD database with 61 new ΔMs, significantly reducing the number of unknowns in the mouse liver phosphoproteome (Figure 1F) and simplifying result interpretation.

### 2.2. OS-PTM Uncovers Biologically Relevant Phosphate-Containing PTMs from Mouse Liver Phosphoproteome

#### 2.2.1. ADP- and Phosphoribosylations

In addition to phosphorylations, OS-PTM reveals in the phosphoproteome other low-abundant biologically relevant modifications due to the presence of phosphate groups in their structure (Appendix A). Thus, we detected abundant phosphoribosylations (PhoRib) (Figure 1E, Appendix A) with ΔM of 212 Da, which may come from ADP-ribosylations either by in-source fragmentation [32] or due to the biological activity of phosphodiesterases in the sample, the enzymes responsible for eliminating ADP, leaving the ribophosphorylated protein [33]. Although ADP-ribosylations occur mainly on Arg due to the specificity of the enzymes catalyzing this reaction [34], they have also been detected in Ser, Lys [35], Asp, Glu, Thr, Trp, His and Cys [36]. In this work the PhoRib modification did not show a clear residue specificity (Figure 1E) and intact ADP-ribosylations (541,061 Da) were only detected in several scans of one peptide from UDP-glucose 4-epimerase (Q8R059) (Appendix A), the key enzyme in the conversion of galactose into the intermediate metabolite of glycolysis glucose 6-phosphate. Thus, it was not possible to reliably establish the modified site, neither of PhoRib modifications nor of ADP-ribosylations, confirming their extremely labile and technically demanding character due to the complexity of their fragmentation pattern [37]. In fact, authors who manage to detect PhoRib or ADP-ribosylations by closed searches do so by extensive manual curation of ADP-ribosylation-specific fragment ions: adenosine (250 Da), adenine (136 Da), AMP (348 Da) and ADP (428 Da) [15]. Although it was not possible to identify the exact ADP-ribosylated residue in UDP-glucose 4-epimerase, the modification was found in the peptide that covers residues 184–195, which contains Y184, the binding site of the cofactor NAD+, and the substrate binding site [38]. In humans, epimerase malfunction leads, by poorly understood mechanisms, to the complex disorder known as galactosemia [39]. ADP-ribosylation is a completely new modification of this enzyme and appears to be in a functionally important region.

#### 2.2.2. Glycerophosphorylations on Lys

We also detected abundant glycerophosphorylations (167.98 Da) on Lys (pgK) (Figure 1E, Appendix A). This PTM in proteins is not enzymatically catalyzed. It is based on the electrophilicity of the glycolytic metabolite glyceraldehyde-3-phosphate produced by Gapdh [2]. It is a bulky modification that switches Lys charge from positive to negative. Thus, by applying an additional stringent filter, considering that all pgK peptides must be partial digestions, we identified 205 proteins and 419 pgK sites (Appendix A), greatly expanding the number of known pgK sites in this tissue [2]. The most glycerophosphorylated protein in the liver is Eno1 (P17182) (Figure 2A), a glycolytic enzyme that catalyzes the conversion of 2-phosphoglycerate to phosphoenolpyruvate, and participates in cell growth control, tolerance to hypoxia and allergic responses. Its most abundant pgK site is K343 (Appendix A), the enzyme active site. Phosphoglycerol binding to K343 inhibits Eno1 activity, reducing the flow through glycolysis to pyruvate [2]. PgK343 is followed in abundance by pgK193, to our knowledge a novel glycerophosphorylated site (Appendix A), also known as the succinylation site mediated by the metabolic regulator SIRT5 [40]. This overlap of two different modifications, derived from primary metabolites of glycolysis on the same residue with proven functional importance, suggests a coregulation between glycerophosphorylation and succinylation in the control of energy homeostasis in the liver.

Figure 2B demonstrates pgK-modified proteins impact diverse metabolic pathways but mostly glycolysis, and span almost all subcellular compartments. Our results suggest this glycolytic metabolite-derived PTM should be a relevant regulatory mechanism for a diverse spectrum of cellular proteins in multiple cellular compartments.

#### 2.2.3. Glycerylphosphorylethanolamine Modifications on Glu

For the first time in a phosphoproteome, to our knowledge, we also identified glycerylphosphorylethanolamine (197.04 Da) modifications on Glu (gpetE) (Figure 2C, Appendix A). This PTM has so far only been found over 30 years ago [41,42] in Elongation factor 1-alpha 1 (Eef1a1, gpetE301 and gpetE374), a protein with two separate functions: RNA binding during the elongation phase of protein translation and cytoskeletal organization [43]. The affected Glu residues are conserved in eukaryotes and protozoan parasites [44]. The ethanolamine on this PTM is donated by the lipid phosphatidylethanolamine (PE) of the Kennedy pathway by a still unknown mechanism, and it has been proposed that PE-modified Eef1a1 would be temporary membrane-bound, while the removal of the acyl chains from the PE lipid moiety by cellular esterases would render the cytosolic gpetE-Eef1a1 form [45].

Here we present, in Figure 2C, an additional eight proteins with gpetE-modified sites detected with >1 PSM in mouse liver (Appendix A), demonstrating for the first time that this rare PTM is not reduced to Eef1a1, further advancing our understanding of this type of PTM. As expected, the hepatic protein accumulating the highest amount of gpetE sites is Eef1a1 (E301, E297 and E293) (Appendix A), although the gpetE374 site was not detected (Appendix A). However, contrary to our observations with pgK, the other metabolite-derived PTM identified in this work, none of the gpetE-modified proteins were functionally associated and they belonged to different subcellular compartments.

#### 2.2.4. Flavine Mononucleotide (FMN) on His

This analysis discovered a novel presence of abundant fmnH-modified peptides in a phosphoproteome (Figure 2D, Appendix A). The three fmnH-modified proteins (annotated spectra in Appendix A) are flavoenzymes and their fmnH sites identified here are known [46,47,48]. While L-gulonolactone oxidase (Gulo) is a critical enzyme required for VitC biosynthesis, mitochondrial Dimethylglycine dehydrogenase (Dmgdh) and Sarcosine dehydrogenase (Sardh) catalyze consecutive reactions of the choline catabolic pathway (Figure 2E).

#### 2.2.5. Some of the Newly Found Biologically Relevant PTMs Are Responsive to MASLD Development

To test if the newly discovered phosphate-containing PTMs were functionally relevant, we quantified, by a label-free method, the relative differences in modified protein sites identified in our OS-PTM analysis during a publicly available time-course MASLD development experiment [24]. The phosphopeptide-enriched raw MS data corresponded to three animal groups: animals fed a high-fat diet for 3 (HFD3, n = 3) or 12 (HFD12, n = 3) weeks, and control animals fed a standard chow (n = 3). After treatment, all mice were sacrificed at 16 weeks of age. According to the original authors [24], after 3 weeks of HFD mice were normolipidemic, but developed insulin resistance, while after 12 weeks they already had hypertriglyceridemia and fatty liver, so they had steatohepatitis.

To quantify label-free OS-PTM experiments we used standardized spectral counting and the median of all nine samples as reference in the log2ratios calculations. This approach is more conservative than using the mean of the three healthy control samples in the log2 ratio denominator. However, using a less variable reference more robust against outliers, such as the median of a higher number of samples, should provide a more precise estimate of the relative abundances of peptides and proteins [49]. Moreover, we developed a novel workflow to integrate the quantitative information at different levels to increase the statistical power of detecting differences between conditions (Section 4, Appendix A). According to the scheme shown in Appendix A, different modified peptides (*pd*) with the same ΔM (*d*) sharing the same protein (*q*) modified site (*qdna*) on residue type *a* on protein residue number *n* (Ex: partial digestions), were integrated into the same *qdna* element. However, modified peptides sharing the same *qna*, but baring a different *d*, were integrated to a different *qdna* element. For every protein (*q*) the standardized spectral counts (*Zq*) were calculated and a reliable set of 3059 proteins quantified in all the samples—for which the PSM sum across the nine samples was higher than 15 (below that filter were proteins with mainly 1–2 PSM/sample)—was selected for principal component analysis (Appendix A), obtaining an excellent grouping of the replicates within treatment groups and a large separation between groups. Moreover, standardized protein quantitation (*Zq*) of all proteins perfectly followed a normal distribution with zero mean and unit SD (Figure 3B,C). These results indicate our OS-PTM quantitation strategy is reliable.

To automatically detect coordinated protein site responses to HFD, independent of the protein changes, protein-normalized standardized *qdna* quantitations (*Zq_qdna*) were subjected to correlation network clustering, obtaining two clusters (Figure 3A): cluster 1 composed of 457 protein sites decreasing in HFD treatment (Figure 3B), and cluster 2 with 382 protein sites increasing in HFD compared to controls (Figure 3C). The pgK, gpetE and fmnH sites altered by MASLD development in both clusters are shown in Figure 3D,F. The ones downregulated by HFD compared to controls are mainly pgK sites from glycolytic and pentose phosphate proteins (Figure 3E), while the HFD-upregulated sites (Figure 3F) have no direct functional connection (annotated spectra in Appendix A).

#### 2.2.6. Functional Implications of the Newly Found MASLD-Responsive PTMs in Their Respective Proteins

We next examined the crystal structures of the 10 proteins bearing the 13 HFD-changing sites (Figure 3D,F) to evaluate the relevance of these sites to their respective protein function. In Glyceraldehyde 3-phosphate dehydrogenase (Gapdh), a key glycolytic enzyme and the one producing the 1,3-bisphosphoglyceride moiety used for the pgK modifications [2], pgK213 and pgK249 are close to the active site (C150) and the [IL]-x-C-x-x-[DE] target motif for cysteine S-nitrosylation (L243-E248) [50], respectively. The latter motif is responsible for the Gapdh role in translational control of gene expression of inflammation-related mRNAs [51,52,53]. Thus, our results suggest a potential interplay between the protein glycerophosphorylation and S-nitrosylation on Gapdh for inflammation control. Moreover, the sites pgK213 and pgK249 were already reported to be functionally important for Gapdh, reducing enzyme affinity for its substrate [2], while pgK160, a novel glycerophosphorylated-Lys for this protein, is exposed to solvent and its function is unknown (Figure 4A). Phosphoglycerate mutase 1 (Pgam1) is the glycolytic enzyme catalyzing the conversion of 3-phosphoglycerate to 2-phosphoglycerate. Its K100 is the substrate binding site (Figure 4B), at which glycerophosphorylation was also reported [2]. We reason that the doubly negative charge on gpK100 would block substrate binding and hinder Pgam1 activity. Enolase 1 (Eno1) is the enzyme that follows Pgam1 in the glycolytic pathway, converting 2-phosphoglycerate to phosphoenolpyruvate. Glycerophosphorylation of the catalytic K343 (Figure 4D) reduces Eno1 activity [2]. Fructose-bisphosphate aldolase B (Aldob) is another glycolytic enzyme producing 3-phosphoglyceraldehyde from fructose-1,6-bisphosphate. The function of its pgK243 is unknown. However, it is solvent-exposed (Figure 4F) and a target for succinylation and acetylation [54], as is the case with pgK239 (Figure 4C) [54] from Regucalcin (Rgn), a gluconolactonase from the pentose phosphate cycle and critical for VitC biosynthesis, and with pgK180 [55] from 6-phosphogluconolactonase (Pgls) from the pentose phosphate pathway.

The other three HFD-downregulated sites are from two mitochondrial proteins: ADP/ATP translocase 2 (Slc25a5), responsible for adenine nucleotide supply to mitochondria, and Dmgdh (Figure 3D). Slc25a5 K292 is the substrate (ADP) binding site, while K23 faces the internal part of the transporter (Figure 4E), where the substrate binding takes place [56] and is a target for succinylation and acetylation [54]. Glycerophosphorylation of these strategically situated Lys residues would probably perturb translocase substrate binding, with the consequent reduction of mitochondrial adenine nucleotide content. As for Dmgdh, H84 (Figure 4H) is the covalent binding site of its cofactor FAD [46,57]. We are not sure of the implications of Dmgdh activity for a covalent binding of FMN instead of FAD, since covalent flavins are found at the FAD as well as at the FMN level [58,59,60].

As for the HFD-upregulated modified sites, Cytochrome b-c1 complex subunit 6 (Uqcrh) K83 is at the interface between CIII_2_ and CIV (Figure 4I) and is also involved in the interaction with CI in the respirasome [61]. According to [61], when the interaction between CIII_2_ and CIV is more relaxed, the respirasome is in the fully active unlocked conformation. When CIII_2_ and CIV are in tight contact through an extensive interface involving subunit 6, the conformation is locked and inactive, which in turn can contain fully or intermediately assembled CIII_2_. In mouse there is a variable proportion of all these conformations. Moreover, K83 is a target for SIRT-mediated acetylation [62]. Then, K83 modification by a bulky negatively charged group such as phosphorylglycerol would probably favor the locked-to-unlocked respirasome transition and increase energy production efficiency by the electron transport chain.

Concerning the oncogenic Serine-threonine kinase receptor-associated protein (Strap), its gpetE347 is not represented in Figure 4 since the crystal structure of this protein is not yet available. Its Alphafold structure has a very low confidence, and the human Strap sequence differs from mouse in the C-terminal region containing gpetE347. Nevertheless, it should be noted that Strap does not have enzymatic activity and it translocates between nucleus and cytosol by poorly understood mechanisms to regulate multiple biological processes such as RNA splicing, apoptosis and TGFbeta-signaling, among others [63,64,65,66]. Previous data have demonstrated that this C-terminal region is necessary for Strap nuclear translocation and LPS-induced IL6 production in macrophages [67]. Thus, this upregulated lipid-derived PTM in the Strap C-terminal region could act in a way similar to gpetE in Eef1a1 and reflect increased Strap migration from membranes to more soluble fractions upon MASLD development.

**Figure 4 ijms-24-16172-f004:**
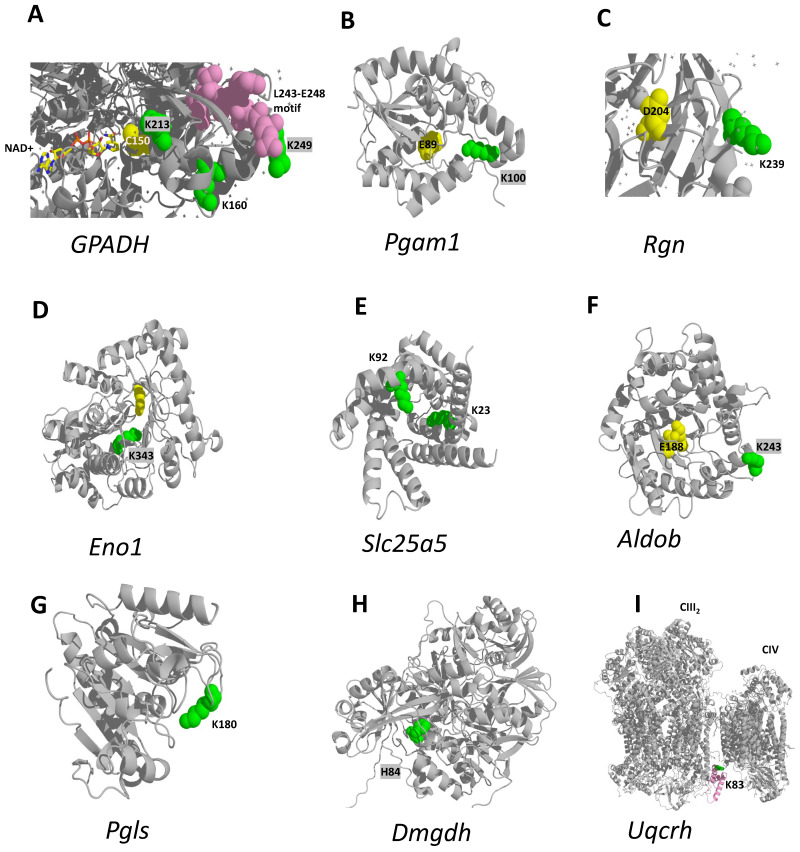
The MASLD-changing pgK- and fmnH-modified sites in mouse liver. (**A**) Gapdh (P16858, PDB 6lgj) pgK213 and pgK249 are close to the active site (C150) and the target motif for cysteine S-nitrosylation (L243-E248) [50], respectively, whereas pgK160 is exposed to solvent; (**B**) Pgam1 (Q9DBJ1, Alphafold) pgK100 is the substrate binding site; (**C**) Rgn (Q64374), PDB 4gn7) pgK239 is solvent-exposed and is a target for succinylation and acetylation [54]; (**D**) Eno1 (P17182, Alphafold) pgK343 is the active site; (**E**) Slc25a5 (P51881, Alphafold) pgK92 is the substrate (ADP) binding site, while pgK23 faces the internal part of the transporter, where the substrate binding takes place [56] and is a target for succinylation and acetylation [54]; (**F**) Aldob (Q91Y97, Alphafold) pgK243 is solvent-exposed and is a target for succinylation and acetylation [54]; (**G**) Pgls (Q9CQ60, Alphafold) is solvent-exposed pgK180 and is a target for acetylation [55]; (**H**) Dmgdh (Q9DBT9, Alphafold) fmnH84 is the FAD binding site; (**I**) Uqcrh (P99028, PDB 7o37) pgK83 is a target for SIRT-mediated acetylation [62]. Strap gpetE347 is not represented since the Alphafold structure has a very low confidence and the human sequence differs in this C-terminal region from mouse. Glycerophosphorylated Lys (pgK) and His modified with flavin mononucleotide (fmnH) are shown in green spheres; active site residues are in yellow spheres; other functionally relevant sites are in pink spheres. Previously reported pgK and fmnH sites are shaded grey.

## 3. Discussion

Phosphoproteome enrichment constitutes a valuable tool to study molecular signaling networks in biological models and is a hot topic in biomedical research. To take full advantage of this tool we need to increase the current knowledge of all PTMs coenriched with phosphopeptides, specially of biologically relevant phosphate-containing PTMs normally not detected in a total lysate due to their low abundance. To increase the current understanding of these types of poorly characterized PTMs we applied the recently developed ultratolerant hypothesis-free proteomics approaches (OS-PTM) to reanalyze a publicly available high-throughput phosphoproteomics experiment in mice in an essential metabolic organ such as the liver [24].

We performed a thorough characterization of the complex modification landscape accompanying IMAC-enriched phosphopeptides. We identified almost 1000 different ΔMs, some with clear residue specificity, on all 20 amino acids, and many residues with different ΔMs. Moreover, a large proportion of the most abundant modifications were unknown. By applying systems-wide approaches we elucidated the chemical and biological nature of an important proportion of unknown modifications. In some cases, and thanks to their peptide terminal specificity, we could at least tell if an unknown modification was introduced during the sample preparation process or was of biological origin, greatly simplifying interpretation of the results. Although, from a biological point of view, these artefactual modifications are of limited interest, they are important for the quality control of a proteomics experiment and to guarantee a high proportion of annotated spectra. Our data reinforce the paramount utility of open searches for PTM study, since such complexity is impossible to foresee in the traditional closed searches, where just a few ΔMs, with their modified residue types, must be set up in advance.

Apart from phosphorylations, we identified the previously reported ADP-ribosylations, phosphoribosylations [15] and Lys-glycerophosphorylations (pgK) [2], from which we obtained an extensive list of hundreds of proteins involved mainly in glycolysis and in many other metabolic pathways from different subcellular compartments and protein complexes. Most of the pgK sites were new, while some were already reported as such or bearing other metabolite-derived modifications (acetylations, succinylations, etc.) involved in metabolic or gene expression control. These findings suggest crosstalk among different metabolite-derived PTMs, such as glycerophosphorylations, acetylations and succinylations, in the control of cellular physiology and provide clues to novel PTMs involved in liver function that are yet to be studied in a functional context.

For the first time in a phosphoproteome, we identified additional metabolite-derived PTMs such as glycerylphosphorylethanolamine-modified Glu (gpetE) and Flavin mononucleotide (FMN)-modified His (fmnH), which we manually validated in the MS/MS spectrum. We greatly expanded the existing knowledge of gpetE sites, since currently the only sites in eukaryotes and protozoan parasites known to have this modification were E301 and E374 from Elongation Factor 1a [44], involved in such a basic cellular function as translation. This rare PTM is derived from the lipid phosphatidylethanolamine during the Kennedy pathway, and it seems to be involved in Eef1a1 translocation between membranes and soluble cellular fractions [45]. Here we provide two additional gpetE sites on Eef1a1, and seven more on other proteins from different subcellular compartments, that to our surprise were not functionally associated. The fmnH-sites on the other hand were localized on two flavo-enzymes catalyzing consecutive reactions in the choline catabolic pathway and were already described. Thus, our data constitute a resource to the scientific community interested in liver protein regulation by PTMs and could be used for hypothesis generation of novel regulations of key proteins.

To assess the functional relevance of the newly discovered phosphate-containing PTMs, we studied them by quantitative OS-PTM proteomics in a global and unbiased manner in the context of metabolic dysfunction-associated steatotic liver disease (MASLD) progression induced by a high fat diet (HFD). MASLD is a complex disorder of great prevalence associated with cancer, cardiovascular and metabolic diseases. The molecular mechanisms behind MASLD progression are under study, little is known about the role of PTMs and there are no approved drugs or non-invasive diagnoses. We demonstrate that specific pgK, gpetE and fmnH levels coming from glycolytic and mitochondrial proteins are responsive to MASLD. Our data, together with the crystal structure analysis of the proteins bearing the HFD-changing sites, suggest that a decrease in the glycolysis-derived pgK modification of catalytic substrate-binding or regulatory residues in most glycolytic enzymes upon HFD-induced MASLD likely increases the catalytic efficiency of these enzymes and glycolysis flow towards pyruvate. Additionally, a decreased pgK modification in the substrate-binding residues from Slc25a5 would potentially increase ADP availability in mitochondria from HFD animals. This, together with a probably more efficient electron transport chain due to the increased glycerophosphorylation of critical Lys residues from the CIII_2_/CIV interface, would promote ATP synthesis that could be used for biosynthetic pathways during MASLD onset. In line with these findings, the original authors of this experiment [24] detect by systems biology analyses a gradual increase in the biosynthesis of unsaturated fatty acids, promoting the characteristic lipid accumulation in the liver upon MASLD development. Our results agree with previous reports of enhanced glycolysis in MASLD due to increased mRNA levels of glycolytic enzymes (HK2 and PKM2) [68] and the augmented liver glucose phosphorylating activity of hexokinase [69]. However, our findings point at new metabolite-derived PTM-based mechanisms behind enhanced glycolysis and fat accumulation in the liver from early insulin-resistant stages of MASLD, which improves our understanding of this complex disease. As for the oncogenic protein Strap, its MASLD-upregulated C-terminal gpetE347 could potentially reflect a promoted translocation from the nucleus to more soluble fractions, with cellular outcomes to be determined.

## 4. Materials and Methods

### 4.1. MS Dataset

The raw mass spectrometry data used in this study (27 .raw files) were obtained from the previously published LC-MS/MS analysis of liver phospho-peptide-enriched fractions from a high-fat diet (HFD)-induced MASLD time-course experiment in mice from the ProteomeXchange repository PXD007653 [24]. Briefly, the experiment contains 3 control male C57BL/6J mice, 3 mice with HFD for 3 weeks starting from 13 weeks of age (HFD3, insulin-resistant, but no fatty liver) and 3 mice with HFD for 12 weeks starting from 4 weeks of age (HFD12, NASH state). Mice were sacrificed at 16 weeks of age. The mouse model and other experiment details can be found in [24].

### 4.2. Data Analysis

Database searching of MS data was performed with MSFragger [18] against a mouse reference proteome concatenated target-decoy database supplemented with pig trypsin and human keratins (Uniport, 18 January 2019, 93,533 sequences). Database searching parameters were as follows: trypsin and LysC digestion with 5 maximum missed cleavage sites, precursor mass tolerance from −230 to 1.500 Da for OS-PTM and 20 ppm for CS, fragment mass tolerance of 20 ppm. Cys carbamidomethylation (+57.021464 Da) was considered to be a fixed modification. For CS, Met oxidation (+15.994915 Da), Ser, Thr and Tyr phosphorylation (+79.966331 Da), as well as protein N-terminal acetylation (+42.010565 Da) were considered to be variable modifications.

Recalibration of experimental *Δmass* values from MSFragger output PSMs, peak modelling and peak assignation were performed as described [19]. For PSM identification 1% FDR threshold was used at global, local (1 Da bins) and *Δmass* peak levels [19] for OS-PTM and 1% FDR for CS. In-house scripts were used to adapt MSFragger output files to the recalibration and validation pipeline from [19].

### 4.3. Statistical Quantitative Analysis

The sum of peptide spectral matches (PSMs) for each element at the protein (*q*) and modified protein-site (*qdna*, where *d* is the delta mass situated on the residue type *a* in position *n* from protein *q*) levels were scaled using total PSM in a particular sample, log2-transformed versus the median from all the samples for the given element and standardized (z-score). Quantitation at the *qdna* level was protein-normalized to detect *qdna* changes independent of their corresponding *q* levels (*Zq_qdna*) (Appendix A). The mapping from modified peptides (*pd*) to protein sites (*qdna*) was performed by in-house scripts. Statistically significant differences between cumulative distributions were assessed using the two-sample Kolmogorov–Smirnov test (**** *p* < 0.0001). Correlation network analysis was performed in Cytoscape 3.9.1 [70] (with the ClusterONE plugin, using Pearson correlation of the quantitative values of all possible protein-normalized modified protein site (*Zq_qdna*) pairs with a correlation coefficient higher than 0.5. ClusterONE settings were minimum size 5, minimum density Auto, Edhe weights correlation, Node penalty 2, Merging method single pass, overlap threshold 0.8. Functional enrichment analysis was performed with STRING (v.12.0) [71] using KEGG, Reactome and GO, with the whole genome as the reference gene set and drawn by Cytoscape. Images of enzyme structures were generated based on PDB or Alphafold files using Pymol (v.2.5.5, http://www.pymol.org, (accessed on 1 July 2023)).

## Figures and Tables

**Figure 1 ijms-24-16172-f001:**
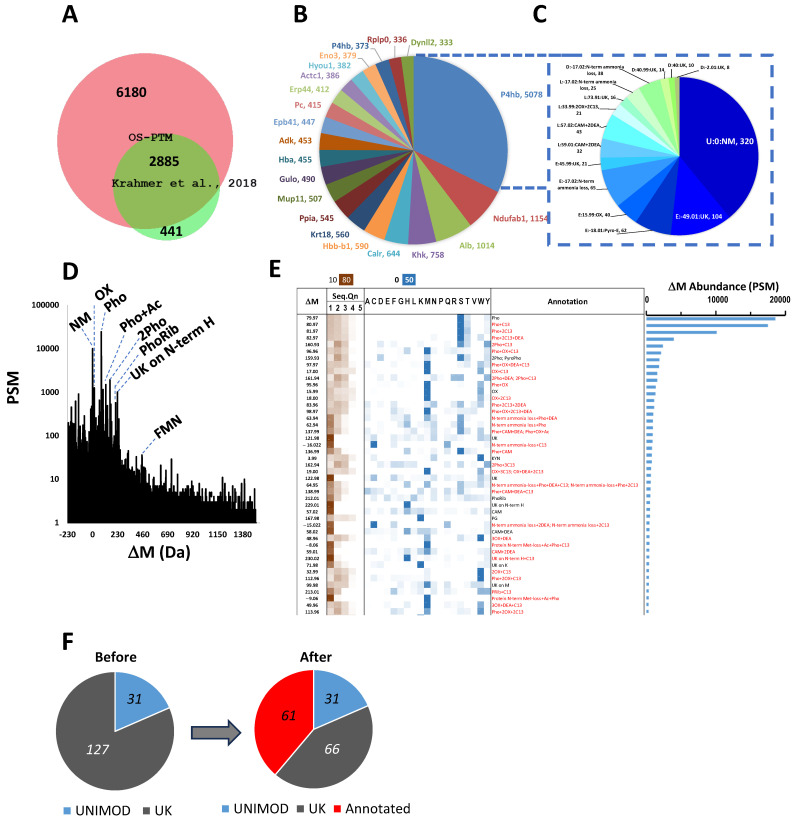
Unbiased proteome-wide open search for post-translational modification (OS-PTM) analysis of mouse liver phosphoproteome reveals a complex picture. (**A**) Area-proportional protein Venn diagrams showing that the OS-PTM approach confirms most originally reported proteins [24], but triples their number; (**B**) top 20 most abundant proteins detected only by OS-PTM. Numbers after the gene name indicate abundance (peptide-spectrum-match, PSMs); (**C**) diagram proportional to the abundance (PSMs) of the top 10 modified residues of Protein disulfide-isomerase (P4hb, P09103). The data are presented in the format *a:d*:*annotation*, *abundance*, where *a* is the type of amino acid and *d* is the delta mass (ΔM) truncated by the second digit after the dot for the sake of clarity. (**D**) Phosphoproteome ΔM distribution. (**E**) The most abundant 50 ΔM peaks responsible for the 77% of the total liver phosphoproteome are residue-specific and span several orders of magnitude in abundance. ΔMs shown are different from 0 (the isotopic envelope of non-modified peptides 1.003, 2.006 and 3.009 Da is considered 0). The ΔM distribution frequency per peptide sequence quintile (Seq.Qn) is expressed as a percentage of the total sum of PSMs for the given ΔM in all the quintiles and is shaded according to the brown color scale at the top. The ΔM distribution frequency per amino acid reside type is expressed as a percentage of the total sum of PSMs for the given ΔM in all the residues after subtracting the average frequency of the three previous and the three subsequent positions and is shaded according to the blue color scale at the top. All unknown ΔMs (not in UNIMOD) that we were able to be elucidated are marked in red in the “Annotation” column. NM: nonmodified; Pho: phosphorylation; DEA: deamidation; CAM: carbamidomethyl; KYN: Kynurein; Ox: oxidation; Ac: acetyl; PhoRib: phosphoribosyl; PG: phosphoglyceryl; UK: Unknown. (**F**) Improvement in unknown ΔM annotation after the system-wide analysis in (**E**). The top 158 ΔM peaks responsible for the 90% of the total ΔM mass were used and 61 new ΔMs were annotated.

**Figure 2 ijms-24-16172-f002:**
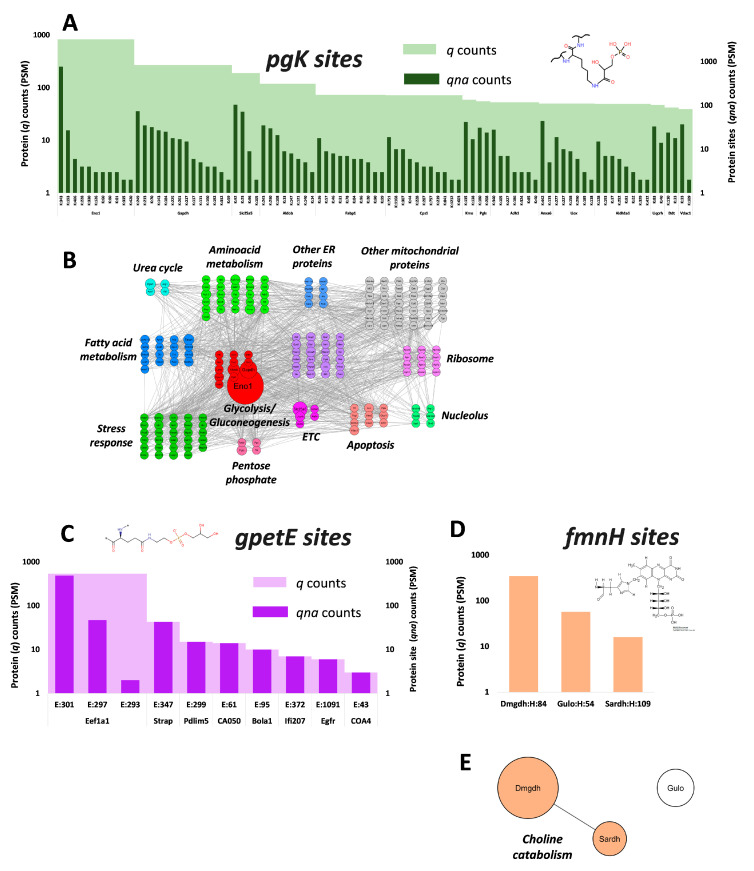
Biologically relevant phosphate-containing PTMs detected in mouse liver phosphoproteome by OS-PTM. (**A**) Top 15 Lys-glycerophosphorylated (167.98 Da) (pgK) proteins with their modified sites (out of 205 proteins and 419 total pgK sites) detected by OS-PTM in the liver phosphoproteome from [24]. Total pgK protein (*q*) and specific pgK protein site (*qna*) PSM counts are shown in light and dark green, respectively. (**B**) Functional distribution of the proteins originating from the pgK sites in mouse liver. Only routes with more than five proteins/route and FDR < 0.001 enrichment are shown. The size of each protein is proportional to its number of pgK modified PSMs. ETC: electron transport chain. (**C**) Proteins modified with glycerylphosphorylethanolamine (GPEt) (197.04 Da) on Glu (gpetE) residues and their modified sites. Only sites with >1 PSM are shown. Total gpetE protein (*q*) and specific gpetE protein site (*qna*) PSM counts are shown in light and dark violet, respectively. * means the rest of the protein. (**D**) Proteins modified with flavin mononucleotide (FMN) (454.08 Da) on His (fmnH) residues and their modified sites. (**E**) Proteins containing fmnH belong to the choline catabolic pathway. All pgK, gpetE and fmnH modified sites are in the format *n:a*, where *n* is the protein residue numbering of the *a* residue type modified by the corresponding mass shift.

**Figure 3 ijms-24-16172-f003:**
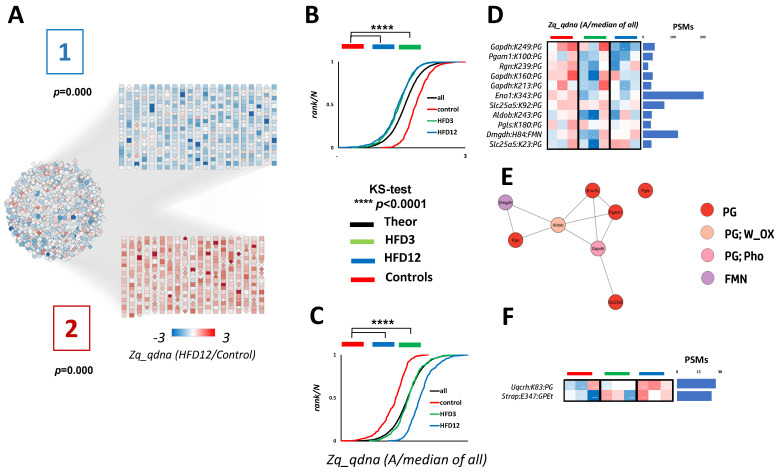
Some of the newly found biologically relevant PTMs are responsive to MASLD development. (**A**) Network clustering of the label-free quantified mouse liver phosphoproteome PTMs. A correlation network analysis using the correlation coefficients of protein-normalized standardized protein site quantifications (*z_q_qdna_*) from all possible protein site pairs in the nine animals (three controls, three HFD3 and three HFD12) was performed using Cytoscape. The list contained only those modified protein sites with a correlation coefficient higher than 0.5, the modification site is known and was quantified in all samples. Cytoscape parameters were min number 5, penalty 2, overlap 0.8. The two statistically significant (*p* < 0.05) non-redundant clusters are presented separated from the bulk of peptides for the sake of clarity with their corresponding *p*-values. The group-averaged *z_q_qdna_* (HFD12/control) magnitude in the heatmap is shaded according to the color scale at the bottom. (**B**,**C**) For each cluster the cumulative distribution of the group-averaged *z_q_qdna_* (A/median of all) values of its composing protein sites is shown (two-sample KS-test *p* < 0.05 against healthy controls), where A is the normalized PSM count in each sample. (**D**,**F**) Heatmap of the newly detected phosphate-containing PTMs contained in Clusters 1 and 2, respectively, associated with MASLD development. All modified sites are in the format *q:an:d*, where *n* is the protein (*q*) and residue numbering of the residue type *a* is modified with the corresponding phosphate-containing PTMs. (**E**) Cluster 1, the new phosphate-containing PTMs decreasing in HFD are concentrated to glycolysis. PG: phosphoglyceryl (167.98 Da); GPEt: glycerylphosphorylethanolamine (197.04 Da); FMN: flavin mononucleotide (454.08 Da).

## Data Availability

The publicly archived raw MS data supporting the reported results can be found at ProteomeXchange (PXD007653).

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
