# Peer review of "Unbiased Phosphoproteome Mining Reveals New Functional Sites of Metabolite-Derived PTMs Involved in MASLD Development"

_ijms, 2023, doi:10.3390/ijms242216172_

Round 1

Reviewer 1 Report

Comments and Suggestions for Authors

Comments on "Unbiased phosphoproteome mining reveals new functional
sites of metabolite-derived PTMs involved in MASLD development." by Moltó et al.

Major comments:

1. Introduction, Line 72-83: Here, authors describe the outline of Open-Search algorithms. However, it is not clear how increasing the mass tolerance for peptide precursors to hundreds of Daltons would affect score and the statistical significance of the identified PTM.

2. Section 2.1, paragraph 1: Here, authors show the output of OS-PTM analysis in Figs. 1A-C. Please substantiate how the data shown in Figs. 1A-C substantiate the statement "These results demonstrate OS-PTM analysis is as reliable as the CS used by the original authors, but much deeper, since it considers all the modifications detectable by MS present in a sample.".

3. Lines 106-107: What do authors mean by 35% of the PTM mass?

4. Fig 1E is difficult to understand. Authors should give proper color bar at the top. What do the blue bars on the right hand side represent?

5. Lines 147-148: "For some unknown delta-Ms we at least managed to discriminate between their artefactual or biological origin." Please explain how.

6. Section 2.2.1: Here, authors describe modifications coming from deltaM of 212 Da. Upto detection it is okay, but "our results point at ADP-ribosylation
as a new potential molecular mechanism for regulating epimerase activity" is a huge extrapolation that lacks evidence.

7. Section 2.2.3: What do authors mean by "the removal of the acyl chains from the PE lipid moiety by cellular esterases would render the cytosolic gpetE-Eef1a1 form."?

8. Lines 239 - 246: Difficult to understand from the limited explanation given for Fig. 2C.

Comments on the Quality of English Language

Line 58: which identification

Lines 61-62: Albeit profound, identifying hundreds of modified sites with a particular PTM, the above-mentioned findings were obtained by traditional closed database searches (CS), .....

Reviewer 2 Report

Comments and Suggestions for Authors

In the manuscript, the authors used OS-PTM to analyze the liver phosphoproteome and identified novel MASLD-associated metabolite-derived PTMs coenriched with phosphopeptides. The work is interesting and provides some new findings regarding metabolic dysfunction-associated steatotic liver disease (MASLD) in mice. However, some additional improvement of the manuscript is required, especially for functional analysis.

Major

In Figure 1A and B authors presented Venn diagrams of reported proteins and phosphoproteome distribution. These results demonstrated a high number of different types of modification of phosphopeptides/phosphoproteins. It would be preferable to present the functional correlation between different types of phospho-PTMs based on KEGG pathways and biological process analysis. To study MASLD, healthy liver should be used as control samples. In the manuscript authors should discuss this issue.

Minor

The figures sometimes are not clear, and it is very difficult to read the text.  The abbreviation in Figure 1E should be delta M and not DM.

For quantitative proteomics analysis, standardized spectral counting was used. What is the reason for that? In general, intensity (area) of precursor ion is more commonly used.

In the section of results, it is mentioned “For every protein (q) the 284 standardized spectral counts (Zq) were calculated and a reliable set of 3059 proteins with 285 >15 PSMs in total and quantified… Do this mean that lower PSMs are not taken in the consideration? It is not clear why.

In Figure 2C,D,E and 3D, there are texts with a lot of abbreviations. What is the meaning of these abbreviations?

Figure 3 describe “Functional implications of the biologically relevant phosphate containing PTMs in MASLD development.” This looks more like network analysis, and it is not clear what is correlation with “functional implication”.

Reviewer 3 Report

Comments and Suggestions for Authors

In this manuscript the authors describe using open searching to reanalyze phosphoproteomics LC-MS data to discover other phospho-containing PTMs.  They reanalyzed a mouse liver phosphoproteomics dataset and found Lys glycerophosphorylation, glycerylphosphorylethanolamine on Glu, and flavin mononucleotide on His.  Some of the PTM abundances correlated with a mouse model of liver disease.  The work is novel and important.  The manuscript is well written.  I have only minor issues.

Minor issues:

The authors should discuss (1-2 sentences) the possibility that the phosphor-enrichment methods used can enrich unmodified (non-phospho) peptides, and peptides with PTMs that do not contain a phospho moiety.

The supplemental figures need to have figure legends.

Line 207: The authors wrote “This PTM [glycerophosphorylation] in proteins is not enzymatically catalyzed.”  They need to describe what is known about the mechanism that causes this PTM (or explain that the mechanism is unknown).

Line 510: The authors wrote “Functional enrichment analysis was performed”.  This analyzes a target gene set against a “reference” (a.k.a. “background”) gene set.  Usually, the reference gene set is the whole genome.  The authors need to indicate what they used for the reference gene set.

Round 2

Reviewer 1 Report

Comments and Suggestions for Authors

Authors have addressed all the queries.